# Feasibility and Early Clinical Experience of Online Adaptive MR-Guided Radiotherapy of Liver Tumors

**DOI:** 10.3390/cancers13071523

**Published:** 2021-03-26

**Authors:** Paul Rogowski, Rieke von Bestenbostel, Franziska Walter, Katrin Straub, Lukas Nierer, Christopher Kurz, Guillaume Landry, Michael Reiner, Christoph Josef Auernhammer, Claus Belka, Maximilian Niyazi, Stefanie Corradini

**Affiliations:** 1Department of Radiation Oncology, University Hospital, LMU Munich, Marchioninistr 15, 81377 Munich, Germany; Rieke.Bestenbostel@med.uni-muenchen.de (R.v.B.); franziska.walter@med.uni-muenchen.de (F.W.); Katrin.Straub@med.uni-muenchen.de (K.S.); Lukas.Nierer@med.uni-muenchen.de (L.N.); Christopher.Kurz@med.uni-muenchen.de (C.K.); Guillaume.Landry@med.uni-muenchen.de (G.L.); Michael.Reiner@med.uni-muenchen.de (M.R.); Claus.Belka@med.uni-muenchen.de (C.B.); Maximilian.Niyazi@med.uni-muenchen.de (M.N.); Stefanie.Corradini@med.uni-muenchen.de (S.C.); 2Department of Internal Medicine 4, University Hospital, LMU Munich, 81377 Munich, Germany; Christoph.Auernhammer@med.uni-muenchen.de; 3Interdisciplinary Center of Neuroendocrine Tumors of the GastroEnteroPancreatic System (GEPNET-KUM, ENETS Centre of Excellence), LMU Munich, 81377 Munich, Germany; 4German Cancer Consortium (DKTK), 81377 Munich, Germany

**Keywords:** MR-guided radiotherapy, MRgRT, oMRgRT, SBRT, MR-linac, liver metastases, cholangiocarcinoma

## Abstract

**Simple Summary:**

Stereotactic body radiotherapy is used in the treatment of liver tumors. However, adjacent organs at risk (OAR) frequently limit the applicable dose to the target volume. The introduction of hybrid magnetic resonance imaging (MRI)-guided radiotherapy systems may allow dose escalation strategies with better OAR sparing due to improved soft tissue visualization, adaptive treatment planning and real-time motion management. Here we report the feasibility and early results of online adaptive MR-guided radiotherapy of primary and secondary liver tumors in eleven patients. The treatment was feasible and successfully completed in all patients. After a median follow-up of five months, no local failure occurred and no ≥ grade 2 toxicity was observed. The technique should be compared to conventional SBRT in further studies to assess the advantages of the technique.

**Abstract:**

Purpose: To assess the feasibility and early results of online adaptive MR-guided radiotherapy (oMRgRT) of liver tumors. Methods: We retrospectively examined consecutive patients with primary or secondary liver lesions treated at our institution using a 0.35T hybrid MR-Linac (Viewray Inc., Mountain View, CA, USA). Online-adaptive treatment planning was used to account for interfractional anatomical changes, and real-time intrafractional motion management using online 2D cine MRI was performed using a respiratory gating approach. Treatment response and toxicity were assessed during follow-up. Results: Eleven patients and a total of 15 lesions were evaluated. Histologies included cholangiocarcinomas and metastases of neuroendocrine tumors, colorectal carcinomas, sarcomas and a gastrointestinal stroma tumor. The median BED_10_ of the PTV prescription doses was 84.4 Gy (range 59.5–112.5 Gy) applied in 3–5 fractions and the mean GTV BED_10_ was in median 147.9 Gy (range 71.7–200.5 Gy). Online plan adaptation was performed in 98% of fractions. The median overall treatment duration was 53 min. The treatment was feasible and successfully completed in all patients. After a median follow-up of five months, no local failure occurred and no ≥ grade two toxicity was observed. OMRgRT resulted in better PTV coverage and fewer OAR constraint violations. Conclusion: Early results of MR-linac based oMRgRT for the primary and secondary liver tumors are promising. The treatment was feasible in all cases and well tolerated with minimal toxicity. The technique should be compared to conventional SBRT in further studies to assess the advantages of the technique.

## 1. Introduction

The incidence of primary and secondary liver tumors has increased over the last decades [1]. In addition to systemic therapies and surgical resection, patients with oligometastatic liver metastases benefit from local ablative therapies such as radiofrequency ablation (RFA), microwave ablation (MWA) or brachytherapy [2,3,4]. Similarly, in primary liver malignancies, i.e., hepatocellular carcinoma (HCC) and cholangiocarcinoma (CCC), several treatment options are used to achieve local control [5,6]. In patients with CCC, resection is the only curative approach, but the majority of tumors are considered irresectable at the time of initial diagnosis. In the palliative setting, chemotherapy with cisplatin and gemcitabine and local ablative therapies such as RFA and Transarterial chemoembolization (TACE) have been shown to be feasible and effective [7].

However, tumor size and location often limit the aforementioned local ablative therapies [8]. Therefore, stereotactic body radiation therapy (SBRT) has gained an increasing role in the treatment of primary and secondary liver tumors [9]. In patients with liver metastases, 2-year local control rates of up to 90% were achieved, depending on tumor size and histologic subtype [10,11,12,13]. Regarding SBRT of CCC, one-year local control rates ranging from 78% to 91% have been reported [14,15]. The local control rate seems to be directly related to the applied dose [15,16,17,18]. However, the tolerance of healthy liver tissue, hepatobiliary structures and adjacent gastrointestinal organs limit the applicable dose. Moreover, inter- and intrafractional mobility of organs at risk (OAR) in the upper abdomen make a precise application even more difficult [19,20]. Therefore, several strategies are used to mitigate and control OAR motion: abdominal compression, respiratory gating and the use of internal target volume (ITV)-concepts [12,14,21,22]. However, none of these approaches allow for an adaptation of the treatment plan to changes in anatomy due to tumor or OAR motion. Furthermore, due to the poor soft tissue contrast in conventional cone beam computed tomography (CBCT), an invasive placement of fiducial markers as surrogate for the tumor position is frequently needed.

The recent introduction of hybrid magnetic resonance image (MRI)-guided radiotherapy systems (MR-linac) into clinical practice represents a new milestone in the field of radiation oncology. These systems have an onboard MRI unit with 0.35–1.5 T integrated with a linear accelerator. The potential advantages include: (1) a better soft tissue contrast than standard radiotherapy CT imaging, (2) less uncertainties through multimodal image registration, (3) daily interfractional online-adaptive treatment planning and (4) real time visualization and intrafractional monitoring of the target. These factors could lead to a more precise target volume and OAR definition. Together with a respiratory gating approach, this may allow for smaller PTV margins and eventually a dose escalation while respecting all dose constraints [23,24,25]. Additionally, MR imaging does not add any ionizing dose exposure to the patient and also does not require invasive implantation of fiducial markers. However, current clinical experience with this new technology is sparse and uncertainties regarding treatment feasibility and safety must be taken into account. The present study reports our initial institutional experience and analyzes the feasibility and early clinical results of primary and secondary liver tumors using online adaptive MR-guided radiotherapy (oMRgRT).

## 2. Materials and Methods

Consecutive patients treated with oMRgRT for primary or secondary liver malignancies at the University Hospital LMU Munich were considered for this analysis and retrospectively analyzed. All patients had a histologically confirmed primary or secondary lesion, were aged 18 years or older and had a Karnofsky performance score ≥ 60. All patients underwent pretreatment imaging with diagnostic MRI, functional imaging with appropriate PET/CT or computed tomography. An interdisciplinary tumor board approved the treatment indication. All analyzed patients were included in a prospective observational clinical trial, which had been approved by the local ethics committee (LMU 20-291). The treatment was conducted using a 0.35 T hybrid MR Linac system (MRIdian, ViewRay Inc., Mountain View, CA, USA).

All patients underwent a MRI simulation scan using true fast imaging with steady state precession (TRUFI)-sequences at the MR Linac. In case of poor visualization of the lesion, gadoxetic acid (Primovist^®®^) was applied as contrast agent. Patients were immobilized in the supine position with arms next to the body or above the head using a dedicated positioning device (MRI Wing step, IT-V, Innsbruck, Austria). The MRI receive coils were placed anterior and posterior to the patient. Simulation imaging was performed in breath-hold (BH), using either an expiration BH or inspiration BH, depending on the optimal patient comfort and where reproducible BH stability was achieved. Thereafter, a standard planning CT using the same patient positioning and BH level was conducted to obtain electron density information. Image datasets were then co-registered using the deformable registration algorithm of the integrated MR Linac treatment planning system. All patients were asked to fast for at least four hours prior to the treatment and were pretreated with scopolamine butylbromide (Buscopan) to decrease bowel motility during imaging and treatment delivery. Moreover, patients were asked to drink a large glass of water (200 mL) 20 min prior to the treatment, as the water increases visibility of the duodenum on TRUFI imaging. Target volumes and OARs were contoured on the 3D MR simulation scan. An isotropic gross tumor volume (GTV) expansion of 3–5 mm was used to generate the planning target volume (PTV).

To account for interfractional anatomical variability, an online-adaptive workflow was applied for all patients, as described previously by Henke and colleagues [26]. Briefly, the treatment plan was superimposed onto the anatomy of the day, and after recontouring of the target volume and OARs, re-optimization was conducted, whenever necessary. During dose delivery, continuous, real-time 2D Cine MRI in a single sagittal plane was used to monitor and limit target volume and OAR motion. For this purpose, a gating boundary contour was defined by adding an isotropic margin of 3–5 mm to the GTV. The beam was gated automatically by the gating functionality of the system through a deformable registration-based tracking algorithm [27]. The maximum percentage of the GTV allowed to be outside the boundary region was set to 3–5%. If this threshold was exceeded, the system automatically stopped the beam. 

For this analysis, treatment and calculated dosimetry data were evaluated. The calculated “delivery time” was derived from the Viewray treatment planning system (TPS) and was defined as the duration of the beam on time, including gantry motion and multi-leaf collimator (MLC) movements of the step-and-shoot IMRT, while “beam on time” was defined as the net beam on duration, as calculated by the TPS. The “overall treatment time” was the duration of the entire fraction as measured by the RTTs from the time of initial imaging to the end of the treatment performed in BH. Follow-up was acquired three months after completion of therapy and every three months subsequently. Toxicity was graded according to the Common Terminology Criteria for Adverse Events (CTCAE) version 5.0. 

To investigate the impact of oMRgRT on target volume coverage and OAR sparing, the non-adapted treatment plans were superimposed on the daily acquired MRI and dose-volume data for target volumes and OARs were calculated and compared with the reoptimized adapted treatment plans. Doses achieved by both plan types for GTV, PTV and OAR for all fractions were retrieved from the TPS workstation and evaluated. For better comparability, dose values were then normalized by constraint for each corresponding structure and patient. The Mann-Whitney U test was used to compare normalized dose values over all patients. 

Data analysis was performed with Excel 2019 (Microsoft Corporation; Redmond, WA, USA) and SPSS (version 26.0; IBM, Armonk, NY, USA).

## 3. Results

Between March and October 2020, eleven patients (six male and five female) with CCC or liver metastases underwent oMRgRT. Median follow-up was five months. Patient and disease characteristics are shown in Table 1. Median age was 66 years (range, 47–86 years) and median Karnofsky performance score was 90% (range, 60–100%). Eight patients had liver metastases, with colorectal adenocarcinoma and neuroendocrine tumors being the most frequent histologies. Two patients had an intrahepatic recurrence of CCC and one patient had a recurrence of a gastrointestinal stroma tumor at the hepatic hilum. Sixty-four percent of patients had undergone prior hepatic local treatments (resection, brachytherapy and/or selective internal radiation therapy (SIRT)).

Treatment characteristics are shown in Table 2. Across all patients, 15 lesions were treated in a total of 47 fractions, with a median PTV prescription dose of BED_10_ 84.4 Gy in 3 to 5 fractions (range 59.5–112.5 Gy). The mean GTV BED_10_ was in median 147.9 Gy (range 71.7–200.5 Gy). In two patients, two lesions were treated simultaneously with the same isocenter, in one patient two lesions were treated simultaneously with two isocenters and in one patient a second hepatic lesion was treated sequentially. The contrast agent Gadoxetic acid was applied in 38.4% of patients. Online-adaptive planning was performed in 46 of 47 fractions (97.8%). The median PTV volume was 39.11 cm^3^ (8.3–411.3 cm^3^) and the median liver volume was 1242.5 cm^3^ (range 879.6–2625.3 cm^3^). The median liver dose was 5.6 Gy (range 2.8–15.5 Gy). Treatment plans were created using a median number of 11 beams (range 7–16), 33 beam segments (range 9–60) and 3367 monitor units per fraction (range 1503–6776), respectively. Examples of treatment plans are shown in Figure 1. The calculated median delivery time from the TPS was 13.07 min (range 6.57–23.41 min) and net median beam-on time was 10.49 min (range 4.03–20.59 min). The overall treatment duration, including the adaptive workflow and treatment delivery using respiratory gating was in median 53 min (range 46–78 min). Unintended treatment interruptions during the application of a fraction occurred in 11 of 47 fractions (23%), but all fractions were completed with no delay within the same treatment session. These interruptions occurred when there was a technical error or when there were difficulties with gating applicability (e.g., the patient could not reach the BH level due to intrafractional motion), which required repositioning of the patient through a couch shift, or when there were difficulties with the anatomical tracking algorithm (e.g., low correlation), which required a manual adjustment of the tracking structure.

The oMRgRT treatment was completed successfully in all patients and was very well tolerated. Overall, six patients reported mild acute toxicity, of which six patients reported CTCAE grade one nausea, two patients vomiting grade one, two patients fatigue grade one and one patient diarrhea grade one. No CTCAE grade two toxicity was observed. One patient died shortly after completion of radiotherapy of a metastasis of a colorectal carcinoma in liver segment VII due to a peritonitis as a result of a duodenal perforation. Radiation plans were retrospectively evaluated and a relationship with the irradiation could be excluded, as the localization of the perforation was not close or within the irradiation area of the treatment (see Figure 1E,F). However, this patient had a prior systemic treatment with bevacizumab. For six patients data was available to calculate pre- and post-treatment Child-Pugh-Score. The value was unchanged in all patients except the one who developed the peritonitis.

All patients received follow-up imaging. There was no local recurrence at the time of this early analysis. One patient treated for a hepatic metastasis of a neuroendocrine tumor developed a new distant hepatic lesion and was treated with a second course of oMRgRT. Further follow-up in this patient showed again a distant hepatic progression and therefore, systemic therapy was changed. One patient treated for a metastasis of a colon cancer progressed with multiple new hepatic lesions. Subsequently, systemic therapy was initiated.

The results of the comparison of non-adaptive plans with adapted plans are shown in Figure 2 and Figure 3. Online adaptation significantly improved PTV coverage. Median PTV coverage normalized to the prescription was 100.2% with oMRgRT vs. 90.3% in non-adaptive plans (*p* < 0.0001). With adaptive treatment planning, none of the fractions missed PTV coverage by >10% compared to 22 fractions (46.8%) with non-adaptive planning. The impact of oMRgRT on compliance with OAR constraint was evaluated for several organs at risk: bowel, duodenum and stomach. The bowel constraint was violated in seven non-adaptive fractions (14.9%). Regarding duodenum and stomach, no constraint violation occurred with adaptive planning compared to constraint violations in three non-adaptive fractions (6.4%).

## 4. Discussion

SBRT of liver tumors is challenging due to frequently poor tumor visibility on CT and on-board CBCT imaging and high inter- and intrafractional variability of anatomic structures in the upper abdomen [28]. In the present study, we report our early experiences with oMRgRT using an online adaptive planning workflow. Our results regarding outcome and toxicity are largely consistent with previous reports.

Feldman et al. analyzed 29 patients with primary and secondary liver tumors treated on a MRIdian Linac system [29]. Patients received 45–50 Gy in 5 fractions (BED_10_: 85.5–100 Gy) or 27–30 Gy in three fractions (BED_10_: 51.3–60 Gy). Mean liver dose was 5.56 Gy, which is consistent with the present cohort. However, only one patient was treated with online-adaptive treatment planning. Similarly, Rosenberg et al. treated 26 patients with primary and secondary liver tumors with a median dose of 50 Gy in five fractions (BED_10_: 100 Gy) [1]. The median PTV size of 98 cm^3^ was much larger as in the present study, and resulted in a more elevated mean liver dose of 21.9 Gy. Gastrointestinal grade three toxicities were reported in 7.7% of cases. Two patients had a drop in Child-Pugh score, which the authors associated with large-volume tumors leading to higher than average mean liver doses. However, in contrast to the present study, oMRgRT was not performed.

After a median follow-up of five months, we found no local failure and the treatment was very well tolerated on early follow-up. Even in the patient in whom a large target volume was irradiated (GTV 318 cc) no side effects occurred. The promising low toxicity in our study is likely due to several factors, including a good target visualization before and during treatment and the routine use of online-adaptive planning with respiratory gated treatment delivery.

Online-adaptive treatment planning aims to compensate for interfractional changes in anatomy [30]. Particularly in the upper abdomen, tumor and OAR positions can vary substantially due to daily changes in filling and distension of the stomach, duodenum and small and large intestines. Henke et al. showed the efficacy of a stereotactic MR-guided online-adaptive radiotherapy (SMART) workflow in a prospective phase 1 trial treating upper gastrointestinal malignancies, including 50% liver tumors [26]. They reported that unintended OAR constraint violations would have occurred in 63% of fractions by sticking to the initial non-adaptive plan, and concluded that the dosimetric benefits of plan adaptation translated to reduced toxicity. In the present cohort, 97.8% of fraction plans were adapted, because the clinical workflow aimed not only for compliance with OAR constraints but also full target volume coverage, even in cases of only minor deviations. In one fraction (2.2%) no adaption was necessary as all constraints and specifications were met.

With oMRgRT, there was a relevant improvement in PTV coverage compared to non-adaptive planning. This is in line with the data of Padgett and colleagues [25], who also compared the two different approaches and demonstrated a deviation from the prescribed dose by more than 10% in 23% of the non-adaptive plans and 0% of the adaptive plans. Additionally, they showed a meaningful sparing of OARs with oMRgRT. This was confirmed in our study in individual cases and especially concerning the OAR bowel.

Due to poor soft tissue contrast in traditional CBCT-based SBRT of liver malignancies, the placement of fiducial markers is frequently used to facilitate image-guidance. Besides the disadvantage of an invasive implantation, this strategy is also associated with other substantial uncertainties, as recently shown by Stick and colleagues [31]. They performed CBCT scans before and after each treatment fraction and intrafractional planar 2D kV scans at every 10 degrees of gantry rotation. The evaluation of differences in fiducials position on pre- and post-treatment CBCTs did not necessarily correspond to the intrafractional motion determined by intrafractional kV imaging. Therefore, the authors concluded that real-time monitoring during treatment delivery, e.g., MRgRT should be preferred. In the present series, we used continuous 2D cine MRI to directly visualize tumor and OAR motion. Respiratory motion management combined with automated beam gating resulted in a high accuracy. An ITV-based approach could be omitted, offering the opportunity of smaller target volumes. This is essential, as the irradiated liver volume correlates directly with post-treatment liver function and is predictive for survival [20]. Therefore, dose limits of 700 cm^3^ receiving <14–18 Gy have previously been suggested [32,33].

In addition to reducing toxicity, minimizing inter- and intra-fractional uncertainties might allow for further dose escalations. This is important as better local control rates have been reported for higher doses in CCC and liver metastases [16,17], and especially in colorectal metastases, which have been reported to be more radioresistant [1,34].

It is worth noting that 40% of the secondary liver lesions in the present analysis were liver metastases of neuroendocrine tumors (NET). In this specific entity, liver-directed surgery is associated with prolonged survival outcomes [35]. Moreover, also local ablative procedures have been shown to be comparably effective [36]. The European Neuroendocrine Tumor Society (ENETS) and the North American Neuroendocrine Tumor Society (NANETS) guidelines recommend an individualized approach depending on age, comorbidities, liver function, number and location of metastases [37,38]. In our small cohort, MR-guided SBRT of NET-metastases was feasible without higher grade toxicities and was associated with adequate early local control rates. However, a longer follow up is needed to draw final conclusions.

Forty-five percent of the patients in our series had prior liver-directed therapies. oMRgRT was feasible without increased toxicity. In one patient with distant hepatic progression, repeated oMRgRT was performed and well tolerated. This is consistent with previous reports [39].

Limitations of the present study are its retrospective nature, the small sample size, the heterogenous cohort and the short follow-up time. Further prospective studies are needed to investigate this promising new irradiation technique.

## 5. Conclusions

In our early clinical experience, online MR-guided radiotherapy enables an effective fiducial-free SBRT of liver malignancies. Daily online-adaptive treatment planning was feasible and tolerated with minimal toxicity. oMRgRT led to better PTV coverage and in individual cases to fewer OAR constraint violations. The technique should be compared to conventional SBRT in further studies to assess the advantages of the technique and facilitate proper patient selection.

## Figures and Tables

**Figure 1 cancers-13-01523-f001:**
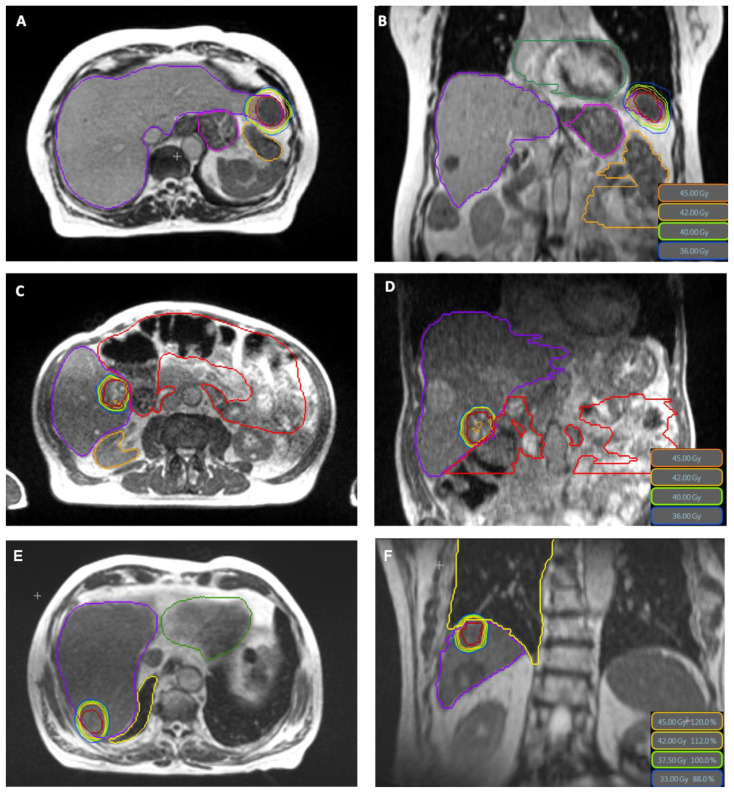
MR-based plans in axial and coronar plane (GTV in red color) with sparing of small bowel (orange), stomach (pink) and heart (green) in (**A**,**B**) and small bowel (red) in (**C**,**D**). (**E**,**F**) show the plan of the patient who suffered a duodenal perforation, with a clear distance between the treated volume and the duodenum. The PTV dose in this case was 37.5 Gy in 3 fractions prescribed to 65% isodose, GTV Mean BED_10_ was 146.7 Gy.

**Figure 2 cancers-13-01523-f002:**
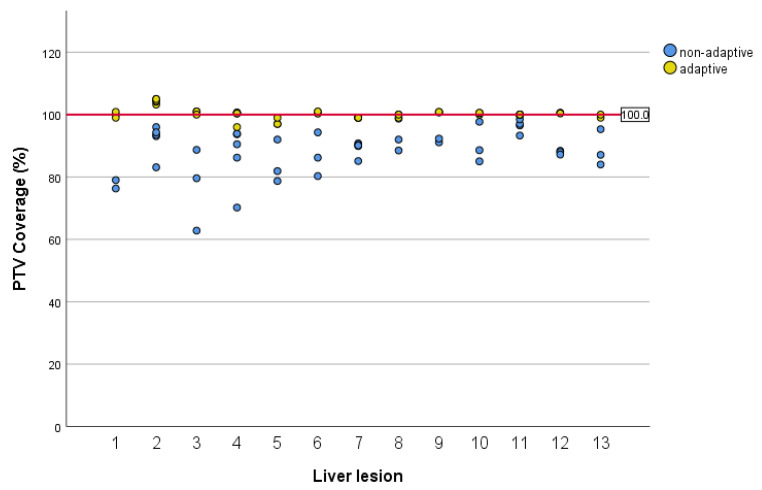
PTV coverage for adaptive and non-adaptive plans. The red horizontal line marks the prescribed level of target coverage. The Y-axis is a relative measure of the PTV coverage. Each dot shows the calculated dose in relation to the prescribed dose.

**Figure 3 cancers-13-01523-f003:**
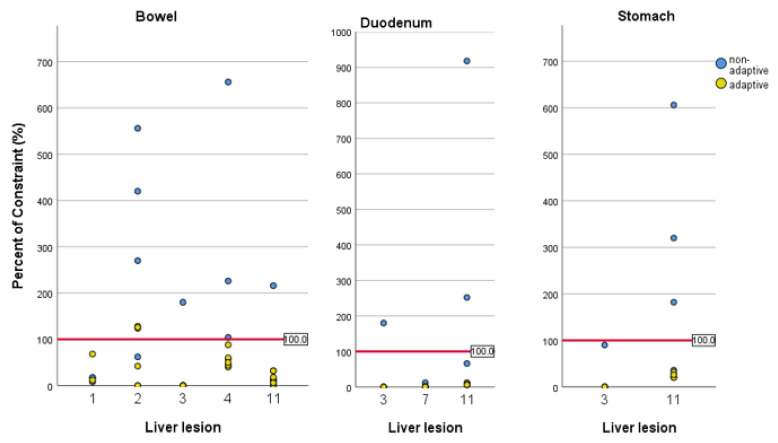
Compliance with OAR constraints for selected treated lesions. For the lesions not shown, there was no constraint violation either with adaptive nor with non-adaptive planning. The red horizontal line marks the prescribed constraint for each patient and OAR.

**Table 1 cancers-13-01523-t001:** Patient characteristics.

Patients, *n*	11
Female, *n* (%)	5 (45.5%)
Male, *n* (%)	6 (54.5%)
**Age (years), median (range)**	66 (47–86)
**Karnofsky performance score, *n* (%)**	
100%	1 (9.1%)
90%	6 (54.5%)
80%	3 (27.3%)
60%	1 (9.1%)
**Histology of treated lesions, *n* (%)**	
Cholangiocarcinoma	2 (13.3%)
Metastasis neuroendocrine tumor	6 (40.0%)
Metastasis colorectal adenocarcinoma	4 (26.6%)
Metastasis sarcoma	2 (13.3%)
Metastasis gastrointestinal stroma tumor	1 (6.6%)
**Pretreatments, *n* (%)**	
Surgery	4 (36.4%)
Brachytherapy	3 (27.3%)
SIRT	1 (9.1%)
No pretreatment	4 (36.4%)

**Table 2 cancers-13-01523-t002:** Treatment characteristics.

Dose Prescription	
15 Gy × 3 (65% isodose, BED_10_: 112.5 Gy)	*n* = 2 (15.4%)
12.5 Gy × 3 (65% isodose, BED_10_: 84.4 Gy)	*n* = 7 (53.8%)
8 Gy × 5 (80% isodose, BED_10_: 72 Gy)	*n* = 2 (15.4%)
7 Gy × 5 (80% isodose, BED_10_: 59.5)	*n* = 2 (15.4%)
PTV Prescription BED_10_ (Gy), median (range)	84.4 (59.5–112.5)
PTV volume (cm^3^), median (range)	39.1 (8.3–411.3)
GTV volume (cm^3^), median (range)	16.5 (1.2–317.8)
GTV Mean BED_10_ (Gy), median (range)	147.9 (71.7–200.5)
Liver volume (cm^3^), median (range)	1242.5 (879.6–2625.3)
Liver Mean dose (Gy), median (range)	5.6 (2.8–15.5)
Number of beams per fraction, median (range)	11 (7–16)
Number of beam segments per fraction, median (range)	33 (9–60)
Monitor units per fraction, median (range)	3367 (1503–6776)
Calculated delivery time (min), median (range)	13 (6–23)
Net beam-on time (min), median (range)	10 (4–20)
Overall treatment duration (min), median (range)	53 (46–78)
Online adaptive planning (number of fractions)	46/47 fractions (97.8%)

## Data Availability

The data presented in this study are available on request from the corresponding author.

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
