# Peer review of "Feasibility and Early Clinical Experience of Online Adaptive MR-Guided Radiotherapy of Liver Tumors"

_cancers, 2021, doi:10.3390/cancers13071523_

Round 1

Reviewer 1 Report

This study presents preliminary results of online adaptive MR-guided Radiotherapy for the treatment of liver tumor in eleven patients. On an average of five months after the treatment, no local failure nor severe toxicity were found. The paper is generally well written, and results seem promising and very interesting.

However, I think that some additional analyses could improve the quality and interest of the work.

Specifically, I suggest the authors to add a comparison with traditional SBRT, at least for DVH of target and OARs, by simulating treatment plans. In this way, it would be possible to highlight the advantages of oMRgRT in terms of constraints coverage in the selected patients, as already shown in Padgett et al [Padgett et al., 2020, Physica Medica, DOI: https://doi.org/10.1016/j.ejmp.2020.07.027]. In fact, due to the limited number of patients, this additional information could offer a deeper insight into the benefits of this technique. For example, the online adaptation has given meaningful reductions in OAR violations with respect to the non-adaptive strategy.

As minor changes, the authors should remove the initial paragraph of Results and Discussion, as it probably remained from template. In addition, the work of Padgett et al. should be mentioned in the introduction, as a work presenting oMRgRT in patients affected by liver tumor.

Reviewer 2 Report

Congratulations for this nicely presented cutting-edge experience. 

As toxicity is one of the main concerns when new technology is applied to patients, I would recommend to expand the information regarding secondary effects observed in this preliminary experience. For instance, a list of toxicity criteria evaluated for this study should be provided and specific liver toxicity parameters, clinical or laboratory, should be communicated if it is possible.

Minor corrections:

Lines 147-149 should be deleted.

Lines 209-212 should be deleted

Line 159 What SIRT stands for?

Line 205 Fig 1. Chang “patient, who” for “patient who”

Round 2

Reviewer 1 Report

I thank the authors for having considered my suggestions and having performed the required additional analyses. I think that the manuscript, in this new form, is improved and it is now complete. I have no more comments.